# Merkel Cells Are Multimodal Sensory Cells: A Review of Study Methods

**DOI:** 10.3390/cells11233827

**Published:** 2022-11-29

**Authors:** Adeline Bataille, Christelle Le Gall, Laurent Misery, Matthieu Talagas

**Affiliations:** 1LIEN—Laboratoire Interactions Epithélium Neurones, Brest University, F-29200 Brest, France; 2Department of Dermatology, Brest University Hospital, F-29200 Brest, France

**Keywords:** Merkel cells, *ex vivo*, *in vivo*, *in vitro*, *in silico*

## Abstract

Merkel cells (MCs) are rare multimodal epidermal sensory cells. Due to their interactions with slowly adapting type 1 (SA1) Aβ low-threshold mechanoreceptor (Aβ-LTMRs) afferents neurons to form Merkel complexes, they are considered to be part of the main tactile terminal organ involved in the light touch sensation. This function has been explored over time by *ex vivo*, *in vivo*, *in vitro*, and *in silico* approaches. *Ex vivo* studies have made it possible to characterize the topography, morphology, and cellular environment of these cells. The interactions of MCs with surrounding cells continue to be studied by *ex vivo* but also in vitro approaches. Indeed, in vitro models have improved the understanding of communication of MCs with other cells present in the skin at the cellular and molecular levels. As for *in vivo* methods, the sensory role of MC complexes can be demonstrated by observing physiological or pathological behavior after genetic modification in mouse models. *In silico* models are emerging and aim to elucidate the sensory coding mechanisms of these complexes. The different methods to study MC complexes presented in this review may allow the investigation of their involvement in other physiological and pathophysiological mechanisms, despite the difficulties in exploring these cells, in particular due to their rarity.

## 1. Introduction

Touch is an essential sense in the exploration of the environment, for social interactions, for tactile discrimination, and in other life tasks. In mammals, sensory end organs that convert mechanical stimuli into electrical signals deciphered by the central nervous system (CNS) are present in the skin.

According to the conventional point of view, sensory neurons are the exclusive mechanosensory cells. However, recent work indicates that initial detection involves non-neuronal skin cells that can also transduce mechanical stimuli and then communicate with neighboring sensory neurons [1,2,3]. Merkel cells (MCs), first identified in 1875 by Friedrich Sigmund Merkel and also named “touch cells” [4], belong to this non-neuronal cell category [4]. MCs associate with interactions with slowly adapting type 1 (SA1) Aβ low-threshold mechanoreceptor (Aβ-LTMRs) afferent neurons to form Merkel complexes, which are highly specialized epidermal structures present in both hairy and glabrous skin [5]. These complexes, named touch domes (TDs) in hairy skin, are responsible for gentle touch perception. Both MCs and SA1 Aβ-LTMRs express PIEZO2, a mechanically activated ion channel, and are very sensitive to skin indentation, pressure, hair movement and other tactile stimuli [6,7,8]. Thus, MC complexes ensure the fine discrimination of the texture, shape, and other physical properties of an object [9,10].

MCs and SA1 Aβ-LTMRs act as a two-receptor-site model in which MCs and SA1 Aβ-LTMRs communicate together through synaptic contacts [9,11,12,13]. Of note, emerging evidence indicates that MCs and MC complexes are also involved in mechanical itch and pain perception.

Since their discovery, MCs have been the subject of many studies to discover their origin, morphology, and functions. While clearly involved in physiological and pathophysiological mechanisms, their rarity makes their exploitation and study challenging. In the present review, we describe MCs and MC complexes in both human and animal models, according to *ex vivo*, *in vivo*, *in vitro*, and *in silico* approaches. We first explore how *ex vivo* investigations have contributed and continue to contribute to understanding their distribution, structure, and interactions with surrounding cells before presenting the major contributions of *in vivo* studies in the comprehension of their roles and mechanotransduction mechanisms. Third, we appraise the contribution of *in vitro* models and conclude by evaluating the perspectives offered by computational models. These findings suggest the consideration of MCs as multimodal sensory cells and have uncovered previously unsuspected functions for these cells beyond tactile perception, such as involvement in the generation of mechanical pain and itch.

## 2. *Ex Vivo* Studies (Skin Biopsies)

*Ex vivo* studies conducted in human and animal skin samples have historically allowed the characterization of the topography, morphology, and cellular environment of MCs. These methods remain relevant in analyzing the functions of MCs and their interactions with surrounding cells, such as keratinocytes and various subtypes of nerve fibers (Figure 1).

### 2.1. Histology

MCs were first identified in 1875 and called “touch cells” by Friedrich Sigmund Merkel, who used silver staining after osmium fixation on human foot epidermis [4,14]. Afterwards, MCs were found in numerous other species, revealing that MCs are mainly found in the basal layer of the epidermis in skin and in some parts of the mucosa of all vertebrates, including fishes, amphibians, reptiles, birds, and mammals [15,16,17,18]. Due to its high resolution, transmission electron microscopy has been an essential method for studies of MCs [19].

In mammals, MCs are found in hairy skin (mammalian whisker follicles (vibrissae)), glabrous skin (fingertips, footpads) and some mucosal epithelia (taste buds, anal canal, labial epithelium, and palatal mucosa). The distribution and the number of cutaneous MCs vary among species. Located in the basal layer of the epidermis, MCs represent only 0.2 to 5% of the epidermal cell population (0.1% in mouse skin and approximately 1.5% in the human skin) [20]. MCs may be isolated or form clusters in touch-sensitive areas, the so-called TDs in hairy skin [16,21,22,23,24], clusters being the predominant arrangement in mice [25]. In humans, MCs are found at the highest densities in regions involved in tactile perception, such as palms and fingers [26]. In other species, notably in mice, MCs are found in glabrous skin (pads) but also at high densities in hair follicles, particularly at the level of guard hairs and whiskers.

The neural crest was initially thought to be the progenitor pathway for MCs [27], but new evidence strongly suggests another hypothesis. MCs originate from epithelial progenitor cells in the epidermis, just as neighboring keratinocytes [23,28], which they contact through desmosomes [7,29]. Indeed, these progenitor cells express both cytokeratin 17 (CK17), a protein also expressed by keratinocytes of the TD [30,31], and the basic helix–loop–helix transcription factor *Atoh1* (or *Math1*) [23,32]. This transcription factor, necessary for the specification of MCs as demonstrated by *Atoh1*^KO^ mice, is not expressed by other skin cells [32,33,34,35]. MCs are not required for specification or maintenance of the ultrastructure of the TD, as the overlying guard hairs and keratinocytes appear normal in the hairy skin of *Atoh1*^KO^ mice.

MCs synthesize intermediate filaments, which distinguish them in their cytoskeletal structure [30,36]; some of these cytokeratins are effective markers for their identification. CK8 is an early differentiation marker that is present in MCs, although it is preceded by *Atoh1* and *Sox2* [33,37]. CK18 has been highlighted by the team of Moll by immunostaining and electron microscopy [29]. It is localized in the cells of basal keratinocytes; however, the intensity of labeling is stronger in MCs [38]. CK18 and CK19 have been used to distinguish MCs from stem cells, with the latter having only specific labeling for CK19 [39]. Because CK18 is not specific for MCs in the skin, anti-CK20 antibodies are now the most widely used tool to identify them in the skin and in other tissues, such as the oral mucosa [40,41]. CK20 is the most specific marker known for both human and rodent epidermal samples and co-localizes with CK8 in 99% of mouse MCs [32,42]. Consequently, anti-CK20 antibodies offer the highest degree of specificity for MCs, ensuring their straightforward identification whether by light or transmission electron microscopy [14,42,43].

MCs are neuroendocrine cells that are distinguishable from other skin cell types by their ultrastructure. In electron microscopy, the presence of electron-dense neurosecretory granules measuring 80 to 120 nm in diameter opposite the nerve endings located in the dermis has been highlighted [44]. In these granules, specific neuropeptides and biogenic amines are stored and secreted; these granules are necessary to mediate the functions of MCs and are viewable by immunohistochemistry [45]. The observation of the contact zone between the nerve fiber and the MC resembles a synapse-like contact zone by the presence of cytoplasm enriched in mitochondria and clear vesicles as well as a plasma membrane tightly attached to an opposite sensory axon [14]. Synaptic features between MCs and SA1 Aβ nerve fibers are a common structure in different species [46].

MCs express neuronal markers such as protein gene product 9.5 (PGP9.5), various neuropeptides such as met-enkephalin, substance P, and calcitonin-gene-related peptide (CGRP), and classical neurotransmitters such as serotonin (5-HT) [47,48]. The range of molecules differs among species. Thus, the presence of such neuropeptides located in MCs has been shown by immunohistochemistry in different mammalian species [15,49,50,51,52], but is not universal in all of them. For example, met-enkephalin is present in rodents but not in cats, dogs, pigs, or humans [51,52,53]. The neuroendocrine marker CD56, also called neural cell adhesion molecule (NCAM), is present on the surface of human and pig MCs [54,55]. Although only 94% of MCs express CD56 [22], this marker can be used for positive selection of MCs for cell-sorting purposes [41]. In addition to neuropeptides, MCs contain neurotransmitters such as serotonin (5-HT). 5-HT is a typical neuroendocrine marker for the neuroepithelial cells and MCs in fish [56], amphibians [57], and mammals [58]. Mechanotransduction in MCs triggers the release of serotonin present in MC granules, which can be visualized by immunohistochemistry [16]. Adenosine triphosphate (ATP) is another potential neurotransmitter between MCs and neurons that is present in MC vesicles [59,60]. In addition, MCs express vesicular glutamate transporters [61,62].

The synaptic nature of the vesicles has been shown by immunostaining for chromogranin A and synaptophysin [63]. In humans, these two markers are present in mature MCs at 23 weeks of gestational age [64]. The synaptophysin expression pattern is thin, granular, non-confluent, and disposed in the most peripheral part of the cell. Chromogranin expression pattern is also granular and forms a peripheral cytoplasmic ring.

Other identification methods by light microscopy are based on the incorporation of fluorescent FM-dyes, which are water-soluble, lipophilic, styryl, and nontoxic, by living MCs [65]. The fluorescent dyes FM1-43 and AM1-43 are able to stain a variety of cells and sensory tissues. MC complexes can be labeled successfully with AM1-43 after systemic injection [66]. FM1-43 was first discovered as an activity-dependent endocytosis marker that allows ultrastructural localization of dyes within the cell [67]. During the recapture process, this dye is internalized by the cell. Fluorescence from FM1-43 makes it possible to quantify the exocytosis of synaptic vesicles in real time [68]. Similar to fluorescent styryl dyes, quinacrine can also be used to identify MCs [65,69,70]. Quinacrine reacts with purines such as ATP [71]. The choice of fluorescent dye is determined based on the biological effects that must be taken into account. FM1-43 is commonly used for studying neuropeptide secretion and membrane traffic [72] or as a blocker of mechanosensory ion channels [21,73,74]. Quinacrine inhibits certain ion channels and therefore the absorption of Ca^2+^ in neuroendocrine cells [75].

### 2.2. Functionality

For the study of interactions between MCs and neurites, the skin–nerve preparation was developed in 1986; this method is particularly suitable for electrophysiological techniques [76]. In the initial description, a patch of skin with its cutaneous nerve branches attached was excised from the dorsal hind limb of a mature salamander. This was placed on a grid located on a shallow well at the bottom of an infusion chamber. The liquid level in the chamber had been adjusted so that the skin remained barely submerged. In order to identify MCs, administration of quinacrine was performed. This system made it possible to study the synaptic contacts between the MCs and the nerve endings and thus to demonstrate that the nerve terminal itself is the mechanosensory transducer. This type of skin–nerve preparation has been adapted in numerous studies on different species, including bullfrogs, rats, and mice [77,78,79,80,81,82]. In all of these skin–nerve preparations, the epidermis is placed in a perfusion chamber and the nerve is threaded into an adjacent recording chamber for electrophysiological measurements.

Single afferent electrophysiological recordings can be made by connecting a microelectrode to an afferent fiber and synchronizing the tips when the cutaneous sensory end organ is mechanically stimulated [83,84]. Afferent nerve endings of the skin are classified according to their sensitivity to mechanical and thermal stimuli. For example, the typical SA irregular response in primary afferent fibers can be induced by gentle pressure on the dome-shaped structures of the epidermis in hairy skin [85]. The work of Adrian and Zotterman allowed the identification of cutaneous receptors sensitive to low-level mechanical stimuli; these receptors would later be called LTMRs [83,84]. These LTMRs recognize mechanical stimuli such as indentation, vibration, or stretching of the skin or the movement or deflection of hair follicles. LTMRs are divided into somatosensory neuron subtypes distinguished by their distinct sensitivities, conduction velocities (CVs), and adaptation to sustained mechanical stimulation. With the use of the von Frey hair technique, it has been possible to identify the mechanical fields present in skin–nerve preparations [86]. Thus, the somatosensory afferents neurons have been classified by (1) conduction velocity (Aβ fibers, CV ≥ 10 m/s; Aδ fibers, CV < 10 m/s and≥ 1 m/s; C fibers, CV < 1 m/s) and (2) adaptation (rapidly adaptive (RA) and SA). SA touch receptors are indentation detectors that activate continuously during a prolonged stimulus. SA touch receptors can be divided into subtypes 1 and 2. Generally, SA1 responses are found to be associated with MCs [44,85,87]. Feng et al. have demonstrated with skin–nerve preparations that there is an alteration of the static phase of SA1 activation in aged mice devoid of MCs [88]. Consequently, the MC complex (SA1-MCs) has the capacity to distinguish two points in close proximity and thus to indicate the position and the speed of the stimulus, which is indentation of the skin. SA1 receptors show no spontaneous activity [44].

Single afferent electrophysiological recordings have made it possible to demonstrate the involvement of the glutamate receptor in the response of the sinus type I (St I) unit in sinus hair capsules (the equivalent to SA1 in the skin) [89]. This has been demonstrated with the use of kynurenate, an antagonist of the ionotropic glutamate receptor with a broad spectrum, which reliably reduces the responses by St I units. Glutamate released by MCs can be bound by receptors present in MCs (autocrine pathway) [90] or by N-methyl-D-aspartate (NMDA) receptors present in afferent ends of MCs (paracrine pathway) [91]. The involvement of another receptor, the β2 adrenergic receptor (β2AR), present at the afferent end of MCs, indicates that adrenergic signaling acts intrinsically in neurons to activate SA1 responses [78]; this has been demonstrated by direct application of norepinephrine to receptive fields of MC afferents neurons during nerve–skin recordings.

The study of the electrical properties of MCs and their function has been difficult due to their small size, their invisibility in the living state, and their relative inaccessibility. In mice, several teams have worked with postnatal mice during their first hair cycle [92]. MCs, which are epidermal components of gentle touch receptors, have been reported to be more abundant during hair follicle growth [93,94]; the abundance of MCs decreases with age, especially in humans. The use of embryonic or fetal skin has made the study of these cells more approachable [29,38,95]. The use of epidermal leaflets of human embryos *ex vivo* has made it possible to map the three-dimensional distribution of MCs compared to other cell types using microscopy, combined at times with immunostaining. The decrease in density of MCs with age can be explained by their inability to multiply [96]. After the first hair cycle, the cells are more difficult to dissociate and, importantly, the yield is lower [97]. In addition, the tissues covering the MCs are softer and therefore easier to remove to allow access of electrodes for patch clamp studies [98]. Enzymatic and mechanical treatments are necessary to gain access to cells in the epidermis because their location prevents direct electrophysiological recordings using conventional glass microelectrodes in an intact epithelium. However, after treatment, MCs lose their shape and possibly also their function [99]. With the discovery of the involvement of channels in mechanotransduction present in MCs and afferents neurons, electrophysiological studies on the mechanotransduction of MC complexes have progressed.

The mechanosensitive ion channels are encoded by the *Piezo* genes [100]. The use of optogenetics makes it possible to study the specific roles of MCs in mechanosensation [9,101]. Selective optical testing of genetically engineered, light-sensitive MCs has identified patterns of sensory neuron activity that are triggered by activation of MCs. A cationic channel activated by blue light (channel rhodopsin 2, activator) that was introduced by a specific vector into MCs induced action potentials on Aβ sensory fibers, demonstrating for the first time that stimulation of an MC is sufficient to induce a tactile sensory message [9]. Likewise, the use of a proton pump activated by green light (archaerhodopsin-3, inhibitor), allowed the inhibition of MCs during mechanical stimulation and considerably decreased the response of Aβ touch fibers. These results demonstrate that MCs contribute to the sensory fiber response. The mechanosensitive channels of Aβ fibers are therefore not sufficient to produce a normal response. To ensure prolonged depolarization, MCs employ voltage-dependent calcium channels that are activated by the opening of the PIEZO2 channels. It is this mechanism of cooperation between ion channels that ensures a prolonged response and therefore the slow adaptation of the complex to a mechanical stimulus. In electrophysiological studies, for example with patch clamp or calcium imaging techniques, the use of pharmacological agents such as channel inhibitors or activators is essential. Intracellular calcium, an important factor in the sensory transduction of MCs, is mobilized by the process of Ca^2+^-induced Ca^2+^ release (CICR). This effect has been demonstrated in particular by the use of caffeine in electrophysiology on rat hair follicles [102]; caffeine dramatically increased the responsiveness of MC receptors to mechanical stimulation at concentrations that stimulate intracellular release of Ca^2+^. In recent studies, activation of PIEZO2 has been shown to lead to the release of Ca^2+^ in MCs, causing afferent Aβ nerve endings to fire SA1 impulses. Using hair follicles from rat whiskers, Ikeda et al. showed that MCs rather than Aβ afferent nerve endings are the main sites of tactile transduction [13]. In addition, using pharmacological tools, they identified the PIEZO2 ion channel as a mechanical transducer in MCs. Touch-activated currents were attenuated by ruthenium red and gadolinium ions, which are commonly used blockers of mechanosensitive channels. In addition, the application of an antibody directed against an intracellular segment of PIEZO2 considerably reduced mechanically activated currents. These data were confirmed with the injection of lentiviral shRNA *Piezo2* particles into whisker hair follicles to knock down PIEZO2 expression. The involvement of MCs and the PIEZO2 channel in mechanotransduction is well established. PIEZO2 deletion in *Atoh1*^KO^ mice or optogenetic deactivation of MCs produced changes in the phases of SA1 firing. MCs are not required for targeting of SA1 to the skin, but the afferent neurons morphology of SA1, electrophysiological responses, and texture discrimination are all altered in the absence of MCs [9,33,103].

The SA1 Aβ-LTMR afferent neurons and MCs act together to produce biphasic encoding in a two-receptor-site model [9,10]. On the one hand, SA1 Aβ-LTMR transduces the initial dynamic phase of skin indentation. On the other hand, MCs mediate the sustained or static phase in the SA1 Aβ-LTMR through synaptic contacts. MCs transmit tactile signals to their associated afferent terminals via chemical synaptic transmission [9,10,13]. Using a pressure-clamped recording technique on single nerve fibers in afferent nerves of mouse whisker hair follicles, Gu’s team [104,105,106,107] has shown that in MC complexes, synapses are predominantly serotonergic. Hoffman et al. have studied a different part of the mouse body, the cutaneous TD, and have shown that there are adrenergic synapses [78].

These types of skin–nerve preparations have also allowed the study of the sensory capacities of MC complexes. It has been shown that these complexes are endowed with a capacity for mechanoperception [12,13]. It has recently been shown the MCs can also perceive cold. Bouvier et al. have assessed how cooling affects the mechanosensitive role of SA1 receptors by performing isolated cutaneous nerve recordings from wildtype or *trpm8*^KO^ mice [108]. TPRM8 (transient receptor potential melastatin 8) is a channel involved in the detection of cold. These authors measured the activity of SA1 mechanoreceptors in the saphenous nerve during mechanical stimulation of the corresponding receptor field in the skin while maintaining the skin temperature at 30 °C, 22 °C, or 15 °C. After comparing the mean instantaneous frequencies at 30 °C, 22 °C, and 15 °C between wildtype and *trpm8*^KO^ mice, it was found that *trpm8*^KO^ mice exhibited higher frequencies in dynamic and static phases compared to wildtype mice at 22 °C, thus demonstrating SA1 discharge in wildtype mice. Moreover, the temperature of 22 °C did not produce an Aβ nerve fiber spike in wildtype or *trpm8*^KO^ mice in the absence of mechanical stimulus, indicating a lack of thermosensitivity of the nerve terminal itself. These results suggest that the reduction in SA1 firing in response to mechanical stimuli at 22 °C is the consequence of the activation of TRPM8 channels present on MCs.

## 3. *In Vivo* Studies (Mice)

Only *in vivo* studies can demonstrate the sensory role of MCs and MC complexes by exploring physiological or pathological behavior after modification of the genetic pedigree of mice (Figure 1).

### 3.1. Touch Perception

Vibrissae have well-mapped neural circuits and thus an ease of controlling sensory inputs and genetic accessibility. The activity of MC complexes can be studied using behavioral tests. For example, to record peaks of MC afferents neurons during behavior (self-movement or active touch) an optogenetic labeling approach on moving mice has been used [101]. The mice whipped freely in the air and against a pole featured in multiple locations as they ran on a treadmill, generating mechanical signals at the base of the whiskers. These experiments showed that MC and SA responded not only to touch, but also to self-movement.

Until recently, it has been difficult to determine whether SA1 responses are necessary for sensing of pressure or discrimination of shape and texture *in vivo*. With the advent of skin-specific genetic silencing of *Atoh1* and *Piezo2*, the opportunity to test the requirement for MCs in tactile behaviors was made possible. Despite the difficulty of testing the contribution of MC complexes to behaviorally soft tactile responses, recent studies have described behavioral tests to elucidate the role of MCs. In *Atoh1*^KO^ mice, the skin is depleted of MCs [23,33] making responses to mechanical stimulation disappear due to the lack of MC complexes. Female *Atoh1*^KO^ mice show a lack of preference for textured surfaces with their paws but not with their whiskers [103]. These results implicate the MCs of MC complexes in texture discrimination except at the level of the whiskers, in which these effects are probably exerted by other somatosensory afferent fibers. However, the main aspects of tactile sensation remain intact without MC activity [10,103]. *Piezo2*^KO^ mice, containing MCs without PIEZO2 channels (but with PIEZO2 still present in sensory neurons), showed normal responses to most behavioral tests. Using von Frey filament assays, these mice showed a mild deficit in detection of tactile stimuli; thus, the mechanosensitivity of MCs would not be entirely dependent on the PIEZO2 channel. On the other hand, mice lacking PIEZO2 in MCs and in sensory neurons exhibit a profound loss of tactile sensation. To test mechanical and thermal sensitivities, both harmless and harmful, in *Piezo2*^KO^ mice, Ranade et al. performed a battery of behavioral tests [12]. With the von Frey filament technique (static force test), they showed that *Piezo2*^KO^ mice have an impaired ability to respond to high forces, unlike *Piezo2*^WT^ mice, which exhibit a linear increase in filament detection depending on strength. This indicates the role of PIEZO2 in a specific range of mechanical stimuli. With a cotton-swab test, a sweeping motion of the cotton swab under the mouse paw led to consistent withdrawal responses in *Piezo2*^WT^ mice, whereas *Piezo2*^KO^ mice showed markedly reduced responses to the cotton-swab stimulus. These results demonstrate that PIEZO2 is the primary mechanotransducer required for tactile sensation in mammals; this finding was confirmed in the whiskers of *Piezo2*^KO^ rats during a whisker tactile test by Ikeda et al. [13]; the authors tested the sensitivity of the whiskers to determine whether PIEZO2 channels in MCs are required for the behavioral tactile responses of the whiskers. To overcome the issue of innate tactile responses in rats during soft touch (head orientation), a small amount of capsaicin was injected into the facial areas of the rats prior to behavioral testing; capsaicin is known to induce central sensitization, which can amplify behavioral readings, and does not alter the conduction of tactile signals by Aβ afferent fibers. When the hairs of the whiskers were slightly bent to activate MC complexes in rats injected with capsaicin, the animals showed hostile behavior, which was blocked in *Piezo2*^KO^ rats *in vivo*.

### 3.2. Mechanical Itch

Mechanical itch is an unpleasant sensation causing the urge to scratch in response to mechanical stimuli such as light pressure exerted on the surface of the skin, also called mechanically evoked itch (MEI) [109]. This sensation can manifest itself in a chronic, pathological way, especially in the elderly (senile pruritus) or in the context of inflammatory dermatoses (atopic dermatitis, psoriasis) and can greatly alter quality of life. In such pathological contexts, MEI is named alloknesis. Alloknesis, an MEI caused by mild mechanical stimulation, can also occur in this context, which in a physiological situation is nevertheless associated with fine tact.

Feng et al. adapted the well-established von Frey technique to mechanically irritate the skin of young and old mice with different levels of mechanical force [88]. Mechanical stimulation ranging from 0.02 to 0.16 g evoked scratching behavior in old mice in a way not seen in young mice. Next, they reproduced a situation of alloknesis with a well-established acetone–ether–water (AEW) model, which recapitulates the dry itchy rash seen in elderly patients. The AEW-treated *Atoh1*^KO^ mice exhibited improved alloknesis compared to the control mice. In response to mild stimuli by von Frey filaments, these AEW-treated *Atoh1*^KO^ mice showed a significant increase in scratching compared to control mice. However, this was not the case in an imiquimod (IMQ)-induced psoriatic mouse model [110]. To develop a mouse model for psoriasis, IMQ cream was applied to the skin of the rostral back for seven consecutive days. In mice treated with IMQ, the alloknesis score markedly increased compared to mice treated with vehicle only. In addition, through the targeted deactivation of Aβ fibers, Sakai et al. showed that the loss of function of Aβ fibers was sufficient to produce alloknesis. Silencing of Aβ fibers is a strategy dependent on the activity of Aβ fibers by the targeted administration of a membrane-impermeable lidocaine derivative, N-ethyl-lidocaine (QX-314) [111]. The selective entry of QX-314 into Aβ fibers is achieved by the activation of Toll-like receptor 5 (TLR5) with its flagellin ligand. Therefore, either a loss of MCs in the setting of dry skin and aging or a reduction in Aβ fibers in IMQ-induced psoriasis can reduce sustained SA1 afferent neurons firing, resulting in a disinhibition of mechanical itch. Combined, these studies highlight the critical role of complexes of MCs and Aβ fibers in modulating mechanical itch in the skin. Hence, MC complexes would have a role in the inhibition of pruritus induced by mechanical stimulation [88,110,112]. The Aβ fibers of the MC complexes could be the afferent LTMR inhibitor of MEI [88,110]. These *in vivo* findings have identified a new potential therapeutic target for the treatment of alloknesis associated with chronic itching.

In light of advances in understanding of the key factors inhibiting MEI, the identification of the components involved in its initiation still remains a question. However, the activation profiles of LTMRs that lead to the appearance of MEI recorded in healthy men and women readily correspond to those of type C fibers [113]. Fukuoka et al. applied mechanical stimuli (vibrations) to the facial hair of normal healthy human subjects to consistently elicit an itching sensation as intense as that induced by histamine. They then showed that the LTMRs initiating MEI could thus be C-LTMR fibers [114,115], which are present in hairy skin in humans, particularly in the form of epidermal nerve fibers [7,116,117]. Of note, the presence of C and Aδ fibers distributed in TDs, both in humans [118] and in other mammals [44,119,120] whose role remains unknown, has been demonstrated suggesting that they could initiate mechanical itch [121].

### 3.3. Mechanical Pain

Mechanical allodynia is a painful sensation evoked by harmless tactile stimuli. It occurs especially in conditions of injury (nerve damage) or prolonged inflammation, in which even a gentle caress can become painful. Mechanical allodynia is one of the symptoms of clinical pain that manifests in many forms: (1) dynamic allodynia evoked by something as soft as a paintbrush, (2) static allodynia evoked by pressure, and (3) punctate allodynia caused by pinch-like stimuli such as those from von Frey’s filaments.

Mechanical allodynia mechanism involved is similar to that of MEI, including the involvement of PIEZO2 [10,11]. Two main research teams have studied its role in mechanical allodynia. These two teams induced inflammation, either by administration of complete Freund’s adjuvant (CFA) or by injection of capsaicin, in mice whose sensory neurons were *Piezo2*^KO^ [122,123]. CFA is a standard for activating immune responses, producing robust and long-lasting tactile allodynia and thermal sensitization in mice [124], and capsaicin activates nociceptor-specific transient receptor potential cation channel vanilloid 1 (TRPV1) and causes sensitization to peripheral stimuli [125]. Loss of PIEZO2 function in sensory neurons in mice reduced mechanical pain in response to touch after inflammation. These results suggest the participation of PIEZO2 in the detection of mechanical stimuli in the contexts of inflammatory and neuropathic pain [126]. The presence of PIEZO2 in different sites (neuronal and non-neuronal cells of the peripheral nervous system and neurons of the spinal cord and brain), shown in rats among other species, suggests a dual involvement in the detection of external mechanical stimuli in the skin and internal mechanical stimuli in the nervous system or a link to other inter- or intra-cellular signaling pathways in pain circuits [127]. MC complexes partially contribute to the mechanical allodynia produced by peripheral nerve injury in a sex-dependent manner [128].

Because pain is a subjective experience that complicates translation from mouse to human, Szczot et al. performed a quantitative sensory evaluation in a small group of human participants bearing inactivating mutations in PIEZO2 [122]. They used a cream containing capsaicin supplemented with heat on the palm and forearm to cause short-term but intense inflammation. Participants with defects in PIEZO2 were unable to detect soft stimuli (brush, air puff, or vibration) while control participants experienced severe pain. Thus, these results in humans complement those obtained in mice by suggesting that neuronal PIEZO2 is necessary for the development of mechanical allodynia.

In considering the two-site receptor hypothesis, MCs may also have a potential role in converting soft touch into mechanical pain.

## 4. *In Vitro* Studies (Culture of Merkel Cells)

As a supplement to *ex vivo* and *in vivo* approaches, cultures of MCs may be an alternative strategy for studying MCs and improving the understanding of the cellular and molecular mechanisms of MC interactions with the other cells present in skin (Figure 1). However, because of their scarcity in the epidermis and the difficulties of isolating and cultivating these cells, few culture models have been developed to date, with three current options: MCs in monoculture, MCs in contact with other types of cells such as neurons or keratinocytes (co-cultures), and cultures of explants. To date, there are few *in vitro* culture systems for studying MCs and their interactions with sensory afferents neurons; most studies on MCs have involved *ex vivo* analysis [26,129,130].

### 4.1. Monoculture of Merkel Cells

Because of their rarity, collecting sufficient numbers of MCs to perform experiments remains a challenge. In order to study them *in vitro*, teams have carried out an essential step using the purification of green fluorescent protein (GFP)-positive MCs from transgenic mice [10,82,90,92]. These Math1-nGFP mice contain MCs genetically labeled with GFP through Math1 enhancer sequences [131]. All the cells within hairy skin of mice were analyzed through a cytometer; only living GFP positive cells were selected. In these animals, MCs were the only cells detectable by GFP [131]. This technique of purifying MCs from transgenic mice represents a significant advance; however, the method remains restrictive for most laboratories. After purification, cells could be maintained for 2 days in a keratinocyte medium (CnT02) with or without serum, depending on the experiment for which they were intended. These cultures were mainly intended for electrophysiological experiments and not for prolonged *in vitro* studies. Only a few studies have generated monocultures of MCs *in vitro*, and their survival and proliferation has posed a problem. However, one report dating from 1996 highlighted their possible survival for 2 weeks in medium supplemented with serum [132]. To our knowledge, this is the only publication that has reported a monoculture of MCs lasting more than 2 days; indeed, MC monocultures are mainly used during 2 days of culture for electrophysiological experiments [10,92]. The touch sensitivity of MCs has been demonstrated by an essential mechanism of Ca^2+^ signaling via the presence of Ca^2+^ channels at the plasma membrane of MCs [92]. Moreover, these *in vitro* electrophysiological studies have allowed the study of tactile currents mediated by MCs with PIEZO2 channels [10].

After purification, MCs have been used to understand the mechanism of mechanotransduction. Their genetic programming as excitable cells that can participate in touch reception has been verified by DNA chips [90]. Their RNA expression profile was compared with that of other epidermal cells. A total of 362 transcripts, including those encoding neuronal transcription factors, presynaptic molecules, and ion channel subunits, were found to be differentially expressed between MCs and other lineages. The immunoreactivity of MCs for presynaptic proteins and the presence of transcripts of presynaptic molecules in these cells demonstrate the vesicular release potential of the contents of neurosecretory granules of MCs. To determine which voltage-gated Ca^2+^ channels are functional in MCs, Haeberle et al. used a ratiometric Ca^2+^ indicator (fura-2 acetoxymethyl ester) to monitor the cytoplasmic concentration of free Ca^2+^ in MCs with specific antagonists [90]. During depolarization, MCs exhibited massive Ca^2+^ entry into the cell, mainly through two types of channels: L-type and P/Q-type channels [90,92]. The voltage-activated Ca^2+^ channels of the MCs generated Ca^2+^ transients that were amplified by a CICR and could thus invoke synaptic vesicle release. Thus, membrane depolarization in MCs activates a Ca^2+^ signaling cascade that includes voltage-activated Ca^2+^ channels, CICR and Ca^2+^-activated K^+^ channels (BKCa) that can modulate the transduction of mechanical stimuli. Using whole-cell patch clamp electrophysiological recordings performed on MCs extracted from wildtype or *Piezo2*^KO^ mice, the role of PIEZO2 as an essential mechanoreceptor for producing mechanical currents has been demonstrated *in vitro* [10].

### 4.2. Co-Culture of Merkel Cells with Other Cells

Co-culture of MCs with keratinocytes [41] with or without sensory neurons [70,91,133] is more frequent than monoculture of MCs. The first MC culture was performed in 1991 [70]. This team developed a mechanical dissociation technique to extract epidermal cells from newborn-rat whiskers and place them in culture with neurons from sensory or sympathetic ganglia. They showed that the selective innervation of MCs with sensory neurons observed *in vivo* could be maintained *in vitro* by the production of nerve growth factor (NGF) by MCs.

The technique for extracting these epidermal cells has continued to be developed, involving similar steps among different teams despite the use of different species, including rats, mice, hamsters, pigs, and humans [20,21,41,132,133,134]. The epidermis is separated from the dermis by enzymatic digestion (collagenase, thermolysin, or dispase), and then dissociated by trypsin. The cells are recovered after filtration. In most studies, cultures of epidermal cells (keratinocytes and MCs) have been performed at this stage. While some have chosen intraperitoneal injection of quinacrine, the presence of specific neuroendocrine granules and specific marker as CK20 allow MC quantification and differentiation of MCs from keratinocytes in culture by immunolabelling [48,58,70,91,135]. In fact, the presence of serotonin in granules of MCs has been highlighted using *in vitro* culture [58] as the production of NGF by MCs [70].

One major problem in the culture of MCs is their low survival. In 1996, Fukuda et al. conducted monolayer cultures of purified newborn-rat epidermal cells that survived for more than 2 weeks in culture [132], showing that improved culture conditions (adding serum, etc.) could stimulate MC survival [20,70,132]. The need for serum in MC cultures has been validated in particular by a study that showed that after 3 days of culture with serum, MCs took on a dendritic form [41]. The co-culture of epidermal cells with nerve cell lines (NG108-15 or PC12) promoted the survival of MCs, resulting in the hypothesis by Shimohira et al. that nerve cells produce survival or growth factors necessary for MCs [133]. The positive influence of keratinocytes and neurons on MC survival was confirmed by Chateau et al. in a tri-compartmented culture of keratinocytes and MCs from human origin and rat primary neuronal cells [136]. Furthermore, it has been observed *in vitro* by confocal imaging and by ultrastructural studies that there are synaptic-like connections between nerve endings and MCs, reminiscent of *ex vivo* observations [133,136].

From these data, mechanotransduction can be explored *in vitro*. This requires the stimulation of mechanosensitive proteins, the opening of Ca^2+^ channels, and the activation of nerve endings. Boulais et al. separated pig snout MCs and keratinocytes by positive magnetic cell sorting based on CD56 expression to produce cultures enriched with MCs [41]. This type of culture has made it possible to study the proliferative potential of MCs and to study their neuroendocrine properties. *In vitro*, these cells respond to histamine and TRPV4 activation and release vasoactive intestinal polypeptide (VIP), demonstrating an involvement of MCs in cutaneous pathophysiological processes. Their exocytosis of dense core granules was shown to be Ca^2+^-independent, opposite to the Ca^2+^-dependent pathway involved in mechanoreception. By transfecting functional MCs with the cDNA expression construct of Math1, they were also able to explore the responses of MCs to mechanical and osmotic stimuli by electrophysiological methods [21]. Thus, this team demonstrated mechanotransduction properties and showed that MCs act as mechanoreceptors with sustained, force-dependent depolarization elicited by direct mechanical stimulation.

Mechanoreception involving Ca^2+^ signaling in MCs is associated with glutamate-like synaptic release [91]. This claim has been supported using co-culture of MCs isolated from the oral mucosa of hamsters and trigeminal ganglion (TG) neurons isolated from newborn Wistar rats. The increase in Ca^2+^ concentration in trigeminal neurons caused by stimulation of MCs was dependent on the release of glutamate by MCs, which activated NMDA receptors on TG. Moreover, these authors showed that MCs did not release ATP. Thus, the released glutamate activated NMDA receptors on TGs. Synaptic transmission has therefore been established and may underlie mechanosensory transduction through MC complexes.

To study the formation of MC complexes in the whiskers, an organotypic co-culture model was developed [137]. The dynamic process of formation of MC complexes cannot be observed directly due to embryonic development. For this, an *in vitro* model was generated, consisting of a row of whiskers with TG explants from the heads of E12 mice cultivated in type I collagen on a cell culture insert for 7 days. The process was observed under a microscope and analyzed by immunohistochemistry. CK8+ MCs were observed on the 7th day of culture, with neurofilament H+ fibers close to them. The results of this study suggest that the formation of MC complexes can be achieved using this *in vitro* organotypic culture method.

On the border between *in vivo* and *in vitro* studies, the use of engineered skin substitutes (ESSs) *in vitro* has allowed the study of MCs *in vivo* [138,139]. In patients with burned skin, a loss of sensation is observed and may persist over time. Although an innervation deficit has been reported, the recovery of MCs has been sparsely studied. However, recent studies have made it possible to study MCs in ESS grafts transplanted *in vivo* in mice [138]. These ESSs were prepared from fibroblasts and primary human keratinocytes with a biopolymer scaffold for 10 days *in vitro* and then transplanted *in vivo* into mice. The ESS grafts, presenting a hyperproliferative phenotype, exhibited responses similar to those observed in wound healing. The wound-healing phenotype could stimulate the differentiation and proliferation of MCs derived from epidermal progenitors in the ESS. Indeed, it has been shown that in experimental skin lesions, an increased number of MCs were produced; fate mapping showed that their differentiation had been induced by the CK17+ progenitors of the TD [140]. In addition, 4 weeks after grafting, an association of MCs with sensory afferent fibers expressing neurofilament M was shown, and an association with more mature sensory afferent fibers expressing neurofilament H was observed after 8 weeks [138]. Functional studies will be needed to confirm the role of MCs in the recovery of tactile sensation after transplant.

With the same aim to advance the skin regeneration therapies and develop bioengineered skin grafts, Lee et al. reported the possibility to generate skin organoids from human pluripotent cells in which MCs were identified by specific markers [141]. They also showed after >100 days of culture that these skin organoids may be able to form mechanosensitive touch complexes as found in normal skin. As indicated by authors, the molecular and physiological profiling of these neural networks have to be more characterized [141].

Nevertheless, ESS grafts seem to be a promising *in vitro* model, allowing the study of MC complexes.

## 5. *In Silico* Studies (Modeling of Merkel Cell–Neurite Complexes)

Mechanotransduction via MC complexes can also be studied using mathematical tools (Figure 1). Computational modeling can help elucidate the sensory coding mechanisms of MC complexes. Currently, it is not possible to record the sensory encodings that govern tactile function directly from tactile end organs *in vivo*, for instance in mammalian skin. A combined experimental and computational modeling approach may identify simple structural principles that could explain the mechanosensory coding properties of SA1 afferent neurons. *In silico* models generate specific predictions by revealing biological mechanisms for future experimental studies.

In general, mechanosensation is studied at two levels: skin mechanics and neuronal dynamics. Models used before 2014 did not approach a connection of these two levels [142,143,144]; computational models combining these two levels began to appear in 2014 [145,146]. The construction of these two models has been based on three modules: (1) a finite-element model imitating the mechanics of the skin to transform skin displacement into strain energy density at the location of mechanotransduction units, (2) a biphasic transduction module to transform strain energy density values into receptor currents, and (3) a leaky integrate-and-fire model to represent the spike initiation zones. These are the first computational models to have captured tactile encoding by combining skin properties, sensory transduction, and spike initiation.

Lesniak et al. studied the impact of the architecture of MC complexes on responses evoked by touch [146]. They tested the functional consequences of the asymmetric distribution of mechanotransduction sites in SA1 afferents neurons by systematically manipulating the potential configurations of these end organs (variation in number of MC complexes). Little is known about how specific architectural features govern neural firing patterns [147]. The principle of this model is that each MC complex serves as a mechanotransduction unit capable of producing receptor currents and thus initiating spikes. These simulations predicted that the number of transducing units and their arrangement could regulate SA1 afferent neurons firing properties. In these reconstructions, carried out in the study by Lesniak et al. in 2014, the authors observed a wide range of firing properties for SA1 afferents neurons. They showed that in addition to a typical SA1 afferent neuron, a thinly myelinated and unbranched afferent neurons contacts the MCs and could correspond to Aδ fibers, thus corroborating *ex vivo* data.

Gerling’s model has allowed a psychophysical prediction of a range of spatial stimuli. However, relying on physiological data would make the computational models more relevant [145]. In a study published in 2018, the team studied the contribution of MCs and neurites to the generation of mechanically evoked SA1 responses [148]. For this purpose, they built a computational model to synthesize the currents of the individual generators using data from electrophysiological recordings of MCs and dorsal root ganglion neurons obtained *in vitro*. This model predicted that rapidly adapting mechanotransduction currents could not account for tactile coding in MC complexes. It is the first computational model of its kind to have used physiological data to study MC complexes. It is a considerable advantage to be able to rely on parameters of biological origin; however, this model only considers the interaction of an MC with a neuron.

## 6. Conclusions

The MC complexes are complex and rare epidermal sensory end-organs. The data that have been obtained by various approaches are complementary and essential for a better understanding of their functions. Their contribution to gentle touch initiation has recently been deciphered by *in vivo* and *ex vivo* approaches, revealing that they function according to a two-site receptor model in which MCs and Aβ fibers communicate through synaptic contacts, perceiving light mechanical stimuli via the PIEZO2 channel, an essential mechanoreceptor (Figure 2).

*In silico* analyses additionally revealed that Aβ fibers branch according to an asymmetric tree-like pattern on which the specific properties of MC complexes depend. Interestingly, the sensory role of MC complexes is not limited to gentle touch perception. They also contribute to the perception of mechanical allodynia through PIEZO2, with the contribution of MCs which remains to be explored. Similarly, most recent *in vivo* studies have revealed that MC complexes also modulate alloknesis. Additional *in vitro* and *ex vivo* approaches will be useful to identify the respective roles of the different neuronal and non-neuronal components. Thus, as Aδ and C fibers are present within MC complexes, the establishment of a functional co-culture model composed of MCs and Aδ and C fibers, as well as the study of human and rodent skin biopsies searching for potential synaptic contacts between MCs and Aδ and C fibers, could represent the first steps in the identification of MCs as initiators of alloknesis.

## Figures and Tables

**Figure 1 cells-11-03827-f001:**
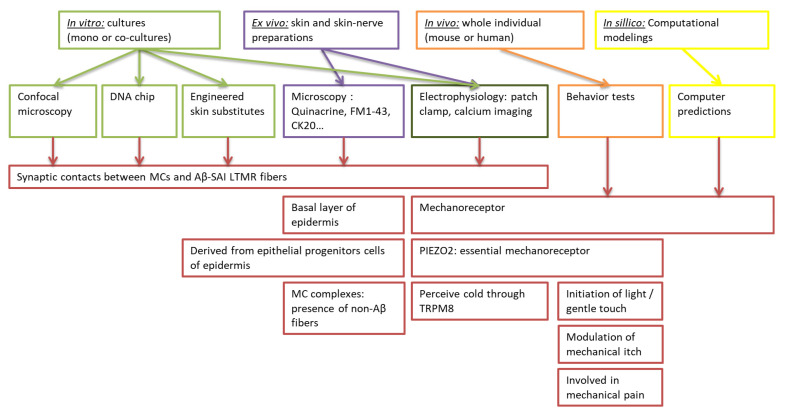
Representative diagram of the different approaches allowing the study of cells that compose the Merkel complexes. The different study methods are *in vitro*, *ex vivo*, *in vivo*, and *in silico*. In each approach, the material and the techniques are different. They complement each other to identify new functions or characteristics attributed to Merkel complexes.

**Figure 2 cells-11-03827-f002:**
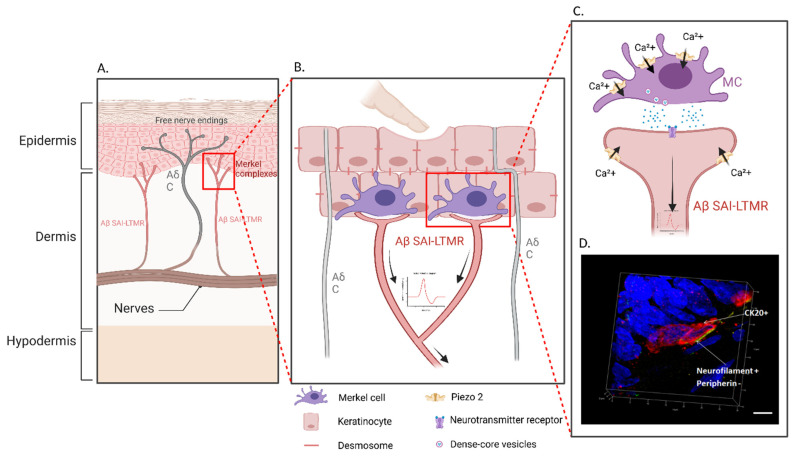
Merkel cell–neurite complexes. (**A**). Merkel cells associated with slowly adapting type 1 (SA1) Aβ-low-threshold mechanoreceptor (Aβ-LTMRs) afferents neurons form the Merkel complex. They are present within the basal layer of the epidermis in both hairy and glabrous skin. (**B**,**C**). A light mechanical stimulation applied to skin triggers action potentials in Aβ SAI-LTMRs, ultimately leading to light touch. Activation of PIEZO2, a mechanically activated ion channel, leads to the release of calcium in Merkel cells, which causes SA1 impulses to be triggered by Aβ afferent neurons. (**D**). Representative three-dimensional reconstruction of a Merkel complex in hairy rat skin. In the photograph, contact between an Aβ fiber (yellow) and a Merkel cell (red) can be observed. Merkel cells were immunoreactive to anti-cytokeratin 20 (red) and the Aβ fiber to anti-neurofilament 200 (yellow) (confocal laser scanning microscopy). Scale bar = 5 µm. (Created with BioRender.com.).

## Data Availability

Not applicable.

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
