# Peer review of "Merkel Cells Are Multimodal Sensory Cells: A Review of Study Methods"

_cells, 2022, doi:10.3390/cells11233827_

Round 1

Reviewer 1 Report

COMMENTS TO AUTHORS

This review article summarizes the findings on Merkel cells from an experimental aspect and is well written. However, there are some concerns as mentioned below.

Major comments:

1.     In vitro studies section: Do you have any findings on Merkel cell lines? Also, do Merkel cells derived from iPS cells exist? If so, please describe them.

2.     Figure 2: Peripheral nerve fibers penetrate within the epidermal keratinocytes, but I do not think this is a common description. Normally, nerve fibers are thought to elongate between cells. What do you think about this?

3.     Conclusion section: The conclusion of this review is too biased toward alloknesis. Please revise the conclusion to be more balanced throughout the entire review.

Minor comments:

1.     There are some parts where there are inconsistencies in the typefaces. Please check them.

2.     Page 5, lines 185-186: What is the unit of conduction velocity? ~ conduction velocity (Abeta fibers, CV  10; Adelta fibers, CV < 10 and  1; C fibers, CV < 1) and ~.

3.     Page 7, line 290:  In vivo => in vivo

4.     Page 11, line 508. What is "VIP" an abbreviation for? ~ TRPV4 activation and release VIP, demonstrating an involvement of ~.

Author Response

Reviewer 1

Dear reviewer,

Thank you for your corrections. We have responded to all of your comments.

 Major comments:

  1. In vitro studies section: Do you have any findings on Merkel cell lines? Also, do Merkel cells derived from iPS cells exist? If so, please describe them.

==>Thank you for this relevant comment. However, we have chosen not to talk about Merkel cell lines because they are Merkel carcinoma lines. We have prefered to address in this review, only the models allowing to study the physiology of the Merkel cell.

Concerning the Merkel cells derived from iPS, we have added a publication referencing the generation of human skin organoids from pluripotent stem cells in which Merkel cells have been identified (from lines 551, reference 144).

  1. Figure 2: Peripheral nerve fibers penetrate within the epidermal keratinocytes, but I do not think this is a common description. Normally, nerve fibers are thought to elongate between cells. What do you think about this?

==>Thank you for this comment. We agree with the reviewer that epidermal fibers elongate between cells, but they also progress within keratinocyte cytoplasmic ensheathments, enwrapped by keratinocyte cytoplasm across their entire circumference, in both the basal and the spinous epidermal layers. (Talagas, M., Lebonvallet, N., Leschiera, R., Elies, P., Marcorelles, P., and Misery, L. (2020b). Intra-epidermal nerve endings progress within keratinocyte cytoplasmic tunnels in normal human skin. Exp. Dermatol. 29, 387–392. https://doi.org/10.1111/exd.14081).

But to schematize this double representation we have modified the figure 2.

  1. Conclusion section: The conclusion of this review is too biased toward alloknesis. Please revise the conclusion to be more balanced throughout the entire review.

==> As recommended by the reviewer we have modified the conclusion.

Minor comments:

  1. There are some parts where there are inconsistencies in the typefaces. Please check them. ==>Corrections have been incorporated into the text in red.
  2. Page 5, lines 185-186: What is the unit of conduction velocity? ~ conduction velocity (Abetafibers, CV ≥ 10; Adelta fibers, CV < 10 and ≥ 1; C fibers, CV < 1) and ~. ==>Corrections have been incorporated into the text in red.
  3. Page 7, line 290: In vivo => in vivo. ==>Correction have been incorporated into the text in red.
  4. Page 11, line 508. What is "VIP" an abbreviation for? ~ TRPV4 activation and release VIP, demonstrating an involvement of ~. ==>Correction have been incorporated into the text in red.

Reviewer 2 Report

To the authors ,

 I am pleased to recommend the manuscript for publications in cells as it is. Only advice the authors to include 2 bibliographic references. 

Line 21. change exploiting for exploring

Line 40. Include in references García-Mesa et al., 2017

Line 221 Include in references García-Piqueras et al., 2019

Author Response

Dear reviewer,

Thank you for your corrections. We have responded to all of your comments.

Comments:

Line 21. change exploiting for exploring. The change have been incorporated into the text in red.

Line 40. Include in references García-Mesa et al., 2017. We have included this reference in text line 40.

Line 221 Include in references García-Piqueras et al., 2019. We have included this reference in text line 215 because we believe it fits the purpose better. Do you confirm this?